# Heurísticas multiarranque para el problema de CMMSA

**Marcos Robles, Sergio Cavero y Eduardo G. Pardo**
Dpto. de Informática y Estadística, Universidad Rey Juan Carlos
C/. Tulipán, s/n, Móstoles, 28933 (Madrid), Spain
{marcos.robles,sergio.cavero,eduardo.pardo}@urjc.es

## Abstract

Un grafo con signos se caracteriza por tener pesos +1 o -1 en sus aristas, representando relaciones positivas o negativas, respectivamente, entre los vértices que unen. Entre otras aplicaciones, estos grafos se emplean en los problemas pertenecientes a la familia de *Sitting Arrangement* (SA), en los que el objetivo es relacionar los vértices de un grafo con signos con los de otro grafo sin signos, evitando que aparezcan relaciones negativas en el camino entre dos vértices conectados positivamente. En este trabajo se aborda una variante del problema del SA, denominada *Cyclic Min-Max Sitting Arrangement*, para la que se proponen tres métodos constructivos y su incorporación en un esquema *Greedy Randomized Adaptive Search Procedure* (GRASP). Los constructivos propuestos se basan en tres criterios voraces distintos: i) la función objetivo; ii) los cliqués de tamaño máximo; y iii) el criterio propuesto por McAllister en [11]. Los métodos son comparados empíricamente con el método que podría considerarse estado del arte para el problema. Los resultados experimentales muestran que la mejor variante GRASP supera sistemáticamente al método previo, en términos de función objetivo y tiempo de ejecución.

## 1. Introducción

Los problemas de embebido de grafos, conocidos como *Graph Layout Problem* (GLP) por sus siglas en inglés [5, 13], son una familia de problemas de optimización combinatoria que tienen como objetivo relacionar los vértices de un grafo de entrada con los vértices de un grafo huésped, de forma que se optimice una función objetivo dada. La primera aplicación práctica de estos problemas fue en el diseño de circuitos, en los que dados unos componentes, se buscaba distribuirlos en una placa de manera que se minimizaran aspectos tales como el cableado utilizado o los cruces entre los cables.

Dentro de los GLP, se encuentra la subfamilia de problemas denotada como *Sitting Arrangement* (SA), que fueron introducidos en 2011 [7]. Estos problemas permiten representar situaciones sociales, indicando relaciones positivas y negativas entre los vértices del grafo, que los hace de gran interés, ya que pueden modelar situaciones que no son reproducibles en otros problemas. Al igual que los demás problemas de GLP, los problemas de SA se definen como una combinación de cuatro elementos: el grafo de entrada, el grafo huésped, una función de embebido y la función objetivo a optimizar.

El grafo de entrada en los problemas de SA se caracteriza por ser un grafo de tipo signo, que es finito, no dirigido y con pesos +1 o -1 en sus aristas. Formalmente, se define como $G = (V_G, E_G, \sigma)$, donde $V_G$ y $E_G$ representan el conjunto de vértices y el conjunto de aristas, respectivamente. La función $\sigma$ está asociada al grafo $G$, asignando a cada arista del grafo una etiqueta o peso +1 o -1, es decir, $\sigma : E_G \to \{-1, +1\}$. Además, se define la función $\Gamma_G(u)$ que, para el vértice $u$, devuelve el conjunto de vértices $v$ para los que existe la arista $\{u, v\} \in E_G$. Esta función se extiende a $\Gamma_G^+(u)$ y $\Gamma_G^-(u)$, las cuales devuelven el conjunto de vértices con los que $u$ forma una arista positiva y negativa,

respectivamente. Estos vértices se denominan "adyacentes positivos" y "adyacentes negativos" a lo largo de este artículo.

La Figura 1a presenta un ejemplo de un grafo de tipo signo $G_1$ con cinco vértices y cinco aristas. El conjunto de vértices está compuesto por $V_{G_1} = \{A, B, C, D, E\}$, y el conjunto de aristas está dado por $E_{G_1} = \{\{A, B\}, \{A, C\}, \{A, D\}, \{A, E\}, \{D, E\}\}$. Para este grafo, la función $\sigma$ devuelve un valor de +1 para las aristas $\{A, C\}$ y $\{A, D\}$, y -1 a las aristas $\{A, B\}$, $\{A, E\}$ y $\{D, E\}$.

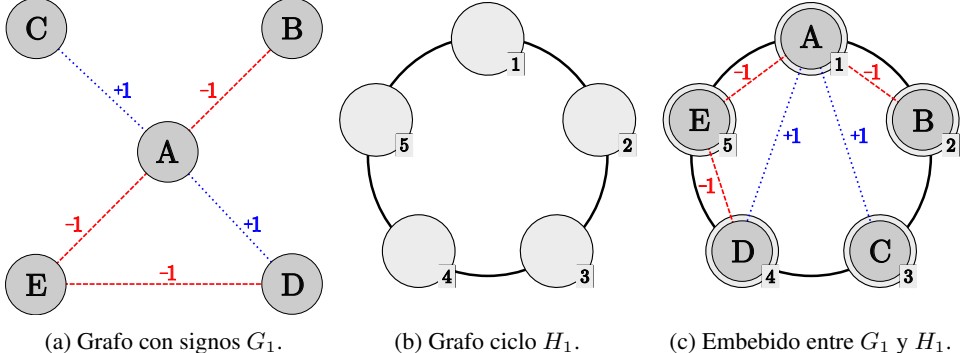

(a) Grafo con signos $G_1$.    (b) Grafo ciclo $H_1$.    (c) Embebido entre $G_1$ y $H_1$.

Figura 1: Representación de un grafo con signos (a), un grafo ciclo (b) y de un posible embebido del primero en el segundo (c).

Originalmente, los problemas de SA se propusieron para un grafo huésped de tipo camino. Sin embargo, este trabajo se centra en el grafo huésped ciclo, dando lugar a la variante del problema conocida como *Cyclic Minimum Sitting Arrangement* (CMinSA). Un grafo ciclo se define como un grafo finito, no dirigido y simple, con una estructura regular donde cada vértice tiene exactamente grado dos y es conexo. Este grafo se define como $H = (V_H, E_H)$, donde $V_H$ y $E_H$ son los conjuntos de vértices y aristas del grafo huésped, respectivamente.

Sobre el grafo huésped, se define la función de caminos $\Psi(i, j)$, tal que, dados dos vértices $i, j \in V_H$, devuelve dos conjuntos de vértices conectados que forman un camino desde $i$ hasta $j$. Nótese que se devuelven dos caminos, ya que al ser un ciclo, para cualquier par de vértices existen dos caminos sin considerar ciclos. Complementariamente, se define también la función $\psi(i, j)$, que selecciona el camino de menor longitud entre esos dos, es decir, aquel compuesto por el menor número de vértices. En el caso especial de que ambos caminos tengan la misma longitud, se elige el que optimice la función objetivo del problema. Si el empate aún persiste, se elige un camino al azar.

La Figura 1b muestra un ejemplo de un grafo de tipo ciclo $H_1$ con cinco vértices. En este caso, el conjunto de vértices está compuesto por $V_{H_1} = \{1, 2, 3, 4, 5\}$ y el de aristas por $E_{H_1} = \{\{1, 2\}, \{2, 3\}, \{3, 4\}, \{4, 5\}, \{5, 1\}\}$. Para ejemplificar el concepto de caminos, se evalúa la función $\Psi(1, 3)$, que da como resultado $(\{1, 2, 3\}, \{1, 5, 4, 3\})$, cada uno correspondiente a un camino posible en el ciclo. Aquí, el primer camino tiene una longitud de tres, mientras que el segundo tiene una longitud de cuatro. Por lo tanto, si se tuviera que evaluar $\psi(1, 3)$, retornaría el primer camino al ser el más corto.

En los GLP se relacionan los vértices del grafo de entrada con los vértices del grafo huésped mediante una función de embebido como $\varphi : v \to i$ donde $v \in V_G$ y $i \in V_H$. Esta función biyectiva es la representación de una solución y, por lo tanto, se evalúa con la función objetivo cuantificando su calidad. El objetivo general de los GLP consiste en encontrar el embebido o solución $\varphi$ entre todos los posibles $\Phi$ que minimice la función objetivo dada. En el problema de CMinSA, la función objetivo cuenta, para cada vértice de entrada $u \in V_G$, cuántos de sus adyacentes positivos $v \in \Gamma^+(u)$ tienen al menos un adyacente negativo $w \in \Gamma^-(u)$ cuyo vértice huésped asignado $\varphi(w)$ se encuentra en el camino más corto $\phi(\varphi(u), \varphi(v))$. Esta situación en concreto se ha denotado en la literatura como "error" y, formalmente, se define el conteo de errores de un vértice como:

$$\epsilon(u, \varphi) = \sum_{v \in \Gamma_G^+(u)} \left| \{w \in \Gamma_G^-(u) : \varphi(w) \in \psi(\varphi(u), \varphi(v))\} \right|$$

donde $\varphi(w) \in \psi(\varphi(u), \varphi(v))$ representa la condición de que el adyacente negativo $w$ se encuentre en el camino al adyacente positivo $v$.

En la variante original de CMinSA, la función objetivo se obtiene sumando los errores individuales $\epsilon(u, \varphi)$. Sin embargo, esta función presenta un inconveniente: la evaluación individual de un vértice puede ser muy elevada, mientras que la del resto es muy baja, y aun así la solución puede ser considerada de alta calidad. Esto tiene el inconveniente práctico de que existen situaciones en las que perjudicar gravemente a un individuo a cambio de beneficiar al resto no es una opción. Para poder tratar esas situaciones, en este trabajo se estudia la variante del *Cyclic Min-Max Sitting Arrangement* (CMMSA), que en lugar de evaluar la función objetivo usando la suma, emplea el máximo, formalizado como:

$$\epsilon(\varphi) = \max_{u \in V_G} \epsilon(u, \varphi).$$

Las definiciones anteriores se ejemplifican usando el embebido $\varphi_1$ mostrado en la Figura 1c, en el que se relacionan $G_1$ y $H_1$. En este caso, los vértices de ambos grafos se relacionan como $\varphi_1(A) = 1$, $\varphi_1(B) = 2$, $\varphi_1(C) = 3$, etc. Para evaluar esta solución, se considera cada vértice de forma independiente. En primer lugar, se evalúa A, que tiene dos adyacentes positivos, C y D, y los caminos más cortos a estos son $\{1, 2, 3\}$ y $\{1, 5, 4\}$, respectivamente. Dado que A tiene adyacentes negativos B y E, asignados en los vértices del grafo huésped 2 y 5, cada uno de estos genera un error. El vértice B lo genera en el camino a C, y el vértice E en el camino a D. Por lo tanto, el vértice A tiene dos errores. El vértice D, que es adyacente positivo de A y negativo de E, también tiene un error, ya que E se encuentra en el camino a A. Los vértices B, C y E no tienen errores, ya que solo tienen adyacentes positivos o negativos. Como resultado, la función objetivo de este embebido es $\epsilon(\varphi_1) = \max\{2, 0, 0, 1, 0\} = 2$.

Esta nueva función objetivo penaliza aquellas situaciones como la descrita anteriormente y favorece soluciones más balanceadas en las que todos los elementos tienen una situación similar. En el aspecto teórico, esta nueva variante cuenta con interés científico, ya que convierte el problema en una situación de minimizar un máximo, con propiedades y aproximaciones específicas, por lo que es de gran interés para esta familia de problemas y para los GLP.

Los principales aportes de este artículo consisten, en primer lugar, en la introducción del problema de CMMSA, que aparece por primera vez. En segundo lugar, se presenta una propuesta algorítmica basada en métodos constructivos heurísticos para este problema y en la metaheurística GRASP. Finalmente, se hace una comparativa de la propuesta algorítmica con el método heurístico empleado en el problema más similar en la literatura, el CMinSA.

El resto del documento se divide en cuatro secciones. En la Sección 2, se desarrollan los trabajos previos que conforman la literatura del problema. En la Sección 3, se formaliza la propuesta algorítmica de este trabajo, explicando los métodos constructivos, la búsqueda local y la metaheurística empleada. Esta propuesta se evalúa experimentalmente en la Sección 4 y se compara con el método del estado del arte del CMinSA adaptado para este problema. Finalmente, en la Sección 5, se presentan las conclusiones y trabajos futuros.

## 2. Trabajos previos

El problema de CMinSA [1] es un problema derivado del problema de *Minimum Sitting Arrangement* (MinSA) [7]. Ambos problemas se diferencian en el grafo huésped: para el CMinSA es un grafo de tipo ciclo [17], mientras que para el MinSA es un grafo de tipo camino [14]. Tanto el CMinSA como el MinSA han sido estudiados desde perspectivas teóricas y prácticas. En particular, los trabajos prácticos se han enfocado en abordar estos problemas utilizando enfoques heurísticos.

Entre los trabajos sobre el MinSA, destacan dos en particular. El primero de ellos es un estudio teórico del problema de 2012 [4], en el que se analizó su complejidad para grafos con una estructura genérica y se demostraba que está fuertemente relacionado con los grafos de tipo intervalo. Esta relación se demostraba al comprobar que, si las aristas positivas del grafo con signos forman un grafo intervalo, se garantiza la existencia de una solución con cero errores. En esencia, esta idea se basaba en agrupar los vértices adyacentes positivos y asignarlos en la solución de forma que se minimice su distancia.

Basándose en la idea de los grafos intervalo, en 2020 se propuso un algoritmo heurístico para el MinSA [15], basado en *Basic Variable Neighborhood Search* (BVNS) [12]. El rasgo distintivo de esta propuesta es su método constructivo, que se fundamentaba en agrupar los vértices adyacentes positivos. Este paso se realizaba en dos fases. En primer lugar, considerando únicamente el subgrafo

formado por las aristas con un peso de +1, se aplica el algoritmo de Bron-Kerbosch [3], el cual devuelve una lista de cliques, donde un clique es un subconjunto de vértices en el que existe una arista entre cada par de vértices. En segundo lugar, estos cliques se asignan de mayor a menor tamaño en la solución. Este enfoque permite obtener soluciones de alta calidad. Sin embargo, presenta el inconveniente de un elevado coste computacional debido a la búsqueda de cliques y al proceso de asignación, en el que se prueban todas las posibles asignaciones para cada candidato seleccionado. Este constructivo se complementaba con un BVNS que emplea una búsqueda local basada en inserciones y una perturbación basada en el movimiento de intercambio.

Centrando la atención en el problema de CMinSA, este fue propuesto originalmente en 2018 [1]. En ese trabajo, los autores realizaron un estudio teórico del problema en el que aplicaron al ciclo los conocimientos del trabajo de 2012 del grafo huésped camino [4]. En este nuevo trabajo demostraron que, si las aristas positivas del grafo con signos forman un grafo circular de arcos, entonces existe una solución con cero errores. En concreto, un grafo circular de arcos es esencialmente un grafo intervalo aplicado en una estructura cíclica. Esto permite, similarmente a MinSA, aprovechar la estructura de grafo circular de arcos para diseñar estrategias que agrupen vértices adyacentes positivos en las soluciones, y explotar este rasgo teórico en propuestas algorítmicas.

En 2023, se presentó una propuesta heurística para el CMinSA basada en BVNS [17]. Específicamente, esta propuesta consistía en una adaptación a la variante del ciclo de un enfoque previo para la variante del camino [15]. Esta adaptación buscaba explotar los rasgos teóricos de los grafos de tipo intervalo, que también son relevantes en el contexto del grafo ciclo. Al existir una equivalencia del grafo intervalo en el grafo ciclo [1], esta propuesta permitió obtener resultados de alta calidad para la variante del grafo ciclo. De forma complementaria, en ese trabajo también se demostraba que el CMinSA y el MinSA presentan rasgos diferentes, aunque la propuesta de BVNS sea efectiva para ambos.

## 3. Propuesta algorítmica

En este trabajo se explora la eficacia de diferentes algoritmos constructivos, trabajando con el esquema multiarranque GRASP [6, 16]. Este esquema se divide en dos partes, una fase de construcción y una fase de mejora. La fase de construcción crea una solución mediante un método constructivo en el que, dado un criterio voraz, aleatoriza su comportamiento en base a un parámetro $\alpha$, diversificando la búsqueda. La fase de mejora aplica una búsqueda local a la solución construida, obteniendo un mínimo local. Estos dos pasos se repiten iterativamente hasta alcanzar un criterio de parada [9].

El esquema general GRASP se presenta en el Algoritmo 1. En cada iteración, se genera una solución inicial utilizando el constructivo semivoraz con el valor de $\alpha$ especificado (Paso 3). Luego, se aplica un procedimiento de búsqueda local a esta solución para mejorarla (Paso 4). Si la solución resultante es mejor que la mejor solución encontrada hasta el momento, se actualiza (Pasos 5-7). En concreto, en este trabajo se proponen los métodos constructivos presentados en la Sección 3.1, y la búsqueda local empleada en la Sección 3.2.

---

**Algorithm 1** *Greedy Randomized Adaptive Search Procedure.*

---

**Require:** $\alpha, maxIter$
1: $\varphi^* \leftarrow$ SoluciónAleatoria()                    ▷ Mejor solución inicial
2: **for** $i \leftarrow 1$ to $maxIter$ **do**
3:     $\varphi' \leftarrow ConstructivoVorazAleatorizado(\alpha, g)$
4:     $\varphi'' \leftarrow BL(\varphi')$
5:     **if** $\epsilon(\varphi'') < \epsilon(\varphi^*)$ **then**
6:         $\varphi^* \leftarrow \varphi''$
7:     **end if**
8: **end for**
9: Devolver $\varphi^*$

---

### 3.1. Métodos constructivos

En este trabajo se presentan tres métodos constructivos específicamente desarrollados para el CMMSA. En concreto, estos métodos constructivos siguen un esquema semivoraz [6], que se presenta

en el Algoritmo 2. Este es un esquema iterativo, en el que se generan múltiples soluciones (Pasos 2-18) y se devuelve la mejor de ellas (Paso 19). Para generar cada una de estas soluciones, en primer lugar, se parte de una solución vacía (Paso 3) sin ningún vértice asignado y se crea una lista de vértices candidatos ($LC$) compuesta generalmente por todos los vértices del grafo de entrada (Paso 4). A continuación, mientras haya candidatos sin asignar y, por lo tanto, sea una solución infactible, se inicia un bucle en el que se construye la solución (Pasos 5-14). En primer lugar, se selecciona un vértice huésped (Paso 6). En un algoritmo voraz, se elegiría el vértice que minimice la función voraz $g$, sin embargo, en los algoritmos semivoraces se selecciona al azar un vértice cuyo valor evaluado esté por debajo de un umbral. En concreto, se define este umbral basado en el valor mínimo y máximo de la función voraz, y el parámetro $\alpha$ (Pasos 7-9). En base a este umbral, se define la lista restringida de candidatos ($LRC$) compuesta únicamente por aquellos candidatos cuya evaluación voraz estén por debajo del umbral (Paso 10). A continuación, se selecciona aleatoriamente un candidato de la $LRC$ (Paso 11), se asigna a la solución (Paso 12), y se elimina de la lista de candidatos (Paso 13). Es importante destacar que, incluso cuando el parámetro de aleatoriedad $\alpha$ se establece a cero, introduciendo un comportamiento aparentemente voraz puro, el algoritmo aún conserva una componente de aleatoriedad. Esto se debe a que, si la $LRC$ contiene múltiples candidatos con el mismo valor voraz mínimo (el umbral en este caso sería igual al valor mínimo), la selección del candidato se realiza de forma aleatoria, permitiendo generar soluciones diferentes en ejecuciones sucesivas, incluso con $\alpha = 0$. Por cada solución generada se evalúa su calidad y, si es la mejor encontrada hasta el momento, se almacena como tal (Pasos 15-17).

---

**Algorithm 2** Algoritmo Semivoraz Multiarranque

---

**Require:** $G, H, maxIter, \alpha, g$

1:   $\varphi^* \leftarrow$ SoluciónAleatoria()                  ▷ Mejor solución inicial
2: **for** 1 to $maxIter$ **do**
3:      $\varphi' \leftarrow$ SoluciónVacía()               ▷ Solución actual vacía
4:      $LC \leftarrow$ ObtenerCandidatos()          ▷ Crear la lista de candidatos
5:      **while** $LC \neq \emptyset$ **do**
6:          $i \leftarrow$ SiguienteVérticeHuésped()         ▷ Elegir vértice huésped
7:          $v_{min} \leftarrow$ mín$\{g(c) \, \forall c \in LC\}$
8:          $v_{max} \leftarrow$ máx$\{g(c) \, \forall c \in LC\}$
9:          $umbral \leftarrow v_{min} + \alpha \cdot (v_{max} - v_{min})$          ▷ Definir umbral
10:        $LRC \leftarrow \{c \in LC : g(c) \leq umbral\}$
11:        $candidato \leftarrow$ SeleccionarAleatoriamente($LRC$)
12:        $\varphi' \leftarrow$ AsignarCandidato($candidato, i$)      ▷ Añadir el candidato a la solución
13:        $LC \leftarrow LC \setminus \{candidato\}$        ▷ Actualizar vértices no asignados
14:      **end while**
15:      **if** $\epsilon(\varphi') < \epsilon(\varphi^*)$ **then**
16:        $\varphi^* \leftarrow \varphi'$
17:      **end if**
18: **end for**
19: Devolver $\varphi^*$

---

En este trabajo se proponen un total de tres criterios voraces $g$. El primer criterio voraz $g_1$ evalúa, para cada candidato, que es un vértice de entrada no asignado $u$, y el vértice huésped actual $i$, el valor de la función objetivo que se obtendría si se asignara $u$ a $i$. Dado que se seleccionará el vértice que minimice la función voraz, se asignará el vértice que minimice la evaluación parcial de la función objetivo.

El segundo criterio voraz es una versión adaptada al CMMSA del constructivo del BVNS propuesto anteriormente en la literatura [17]. En este caso, los candidatos son cliques $Q_G = \{q_1, q_2, \cdots, q_n\}$ generados mediante el algoritmo de Bron–Kerbosch [3], donde $q_i$ es un conjunto de vértices. Este constructivo asigna los candidatos de mayor a menos tamaño, por lo que se define el criterio voraz como $g_2(q_i) = -|q_i|$ y, dado el clique, se asignan sus vértices de forma aleatoria. Siguiendo este criterio, se asignan grupos de vértices adyacentes positivos en la solución de manera que no puede haber adyacentes negativos más cerca que los positivos, evitando así los errores. Dado que el proceso de búsqueda de cliques es costoso, y puede consumir todo el tiempo de ejecución por sí mismo, se ha limitado este paso a tan solo diez segundos al inicio de la ejecución, usando para todas las construcciones únicamente los cliques obtenidos en ese tiempo.

El tercer criterio se ha utilizado en otros problemas similares de tipo GLP [2, 11]. Este criterio se basa en considerar, para cada vértice evaluado, los vértices adyacentes asignados y no asignados, multiplicando cada uno por un peso para cuantificar una medida de "urgencia". Este método se ha adaptado para el problema de CMMSA dividiendo los pesos en adyacentes positivos y negativos, pasando de dos a cuatro pesos. Por un lado, la cuantificación de los vértices adyacentes que ya están asignados se define formalmente como:

$$g_3^{in}(u,\varphi') = w_1 \cdot |\{v_1 \in \Gamma^+(u) : \varphi'(v_1) \neq \emptyset\}| + w_2 \cdot |\{v_2 \in \Gamma^-(u) : \varphi'(v_2) \neq \emptyset\}| \qquad (1)$$

donde $\Gamma^+(u)$ y $\Gamma^-(u)$ son los conjuntos de vértices adyacentes positivos y negativos de $u$ respectivamente, y $\varphi'(v) \neq \emptyset$ indica que el vértice $v$ ya tiene un vértice huésped asignado en la solución parcial $\varphi'$. Por otro lado, la cuantificación de los vértices adyacentes no asignados se define de manera similar:

$$g_3^{out}(u,\varphi') = w_3 \cdot |\{v_3 \in \Gamma^+(u) : \varphi'(v_3) = \emptyset\}| + w_4 \cdot |\{v_4 \in \Gamma^-(u) : \varphi'(v_4) = \emptyset\}| \qquad (2)$$

donde $\varphi'(v) = \emptyset$ indica que el vértice $v$ aún no tiene un vértice huésped asignado en la solución parcial $\varphi'$. Finalmente, se suman ambos valores para obtener el criterio voraz $g_3$:

$$g_3(u,\varphi') = g_3^{in}(u,\varphi') + g_3^{out}(u,\varphi') \qquad (3)$$

Definidos los criterios para seleccionar el vértice de entrada, queda elegir el vértice huésped al que será asignado. En este trabajo, para seleccionarlo se sigue un orden secuencial, en el que partiendo desde un vértice huésped cualquiera se selecciona el siguiente vértice siguiendo el sentido de las agujas del reloj (Paso 6 del Algoritmo 2), siendo este un vértice adyacente al que se ha asignado anteriormente. Tomando el ejemplo de la Figura 1b, el primer vértice huésped a elegir puede ser el 1. A continuación, se elige el 2, el siguiente vértice seleccionado sería el 3, después el 4, y así sucesivamente hasta asignar todos los vértices. Nótese que este método de selección es distinto al propuesto en el trabajo original del constructivo $g_2$ [17] ya que en ese trabajo el clique seleccionado se asignaba en todos los vértices huésped y se asignaba finalmente en el que minimizara la función objetivo. En este trabajo, en su lugar, se prueba solo una opción, convirtiéndolo en un método mucho más ligero, apto para un algoritmo multiarranque.

## 3.2. Búsqueda local

Una búsqueda local es un proceso de mejora que aplica movimientos sistemáticamente a una solución hasta que alcanza un mínimo local. Estos movimientos son específicos del problema para el que se definen, y generalmente, explotan propiedades estructurales del problema. En este trabajo se emplea el movimiento de intercambio (conocido en inglés como *swap*), el cual intercambia las asignaciones de dos vértices sin verse afectado el resto del embebido. Formalmente, dada una solución $\varphi_a$ y dos vértices $u, v \in V_G : u \neq v$ tales que $\varphi_a(u) = i$ y $\varphi_a(v) = j$, al aplicar el movimiento de intercambio la solución resultante $\varphi_b$ cumplirá $\varphi_b(u) = j$ y $\varphi_b(v) = i$. Este movimiento tiene un tamaño de vecindad de $|V_G| \cdot (|V_G| - 1)/2$.

A partir de un movimiento, y una solución de partida, $\varphi$, se puede definir la vecindad de la solución como todas las soluciones que se pueden alcanzar aplicando ese movimiento a $\varphi$. Una búsqueda local explora la vecindad siguiendo una estrategia, que define el comportamiento cuando al aplicar un movimiento se encuentra una solución con un mejor valor de función objetivo. Existen dos estrategias comunes, la primera es conocida en inglés como "*Best Improvement*" y consiste en explorar todo el vecindario buscando la mejor solución del mismo. La segunda estrategia, conocida como "*First Improvement*", realiza el primer movimiento de mejora que encuentre durante la exploración de la vecindad. En ambos casos, si se produce un movimiento de mejora, se reinicia la búsqueda desde la nueva solución. Ambas estrategias son efectivas en diferentes situaciones, por lo que es recomendable analizar en qué caso utilizar cada una. En la experimentación preliminar llevada a cabo se ha observado que la estrategia "*First Improvement*" es más efectiva para este problema, ya que converge más rápido a óptimos locales obteniendo soluciones de similar calidad.

## 4. Experimentación

En esta sección, se presentan los experimentos diseñados para evaluar el rendimiento de los métodos propuestos. En la Sección 4.1 se describen las instancias utilizadas y el subconjunto de estas para

la experimentación preliminar. En la Sección 4.2 se detalla la experimentación preliminar para determinar la mejor configuración de los algoritmos. Finalmente, en la Sección 4.3, se compara la efectividad del algoritmo propuesto con una adaptación directa del algoritmo heurístico empleado para el problema CMinSA en trabajos previos.

## 4.1. Instancias

Para la experimentación en este trabajo se utilizan las instancias disponibles de trabajos relacionados de la literatura [15, 17, 18]. Las instancias se caracterizan por dividirse en tres grupos basados en la estructura interna de los grafos: *Complete*, *Interval* y *Random*.

Las instancias *Complete* tienen una estructura de grafo completo, en el que todos los vértices tienen una arista con todos los demás vértices. En las instancias *Interval*, el subgrafo inducido por las aristas positivas presenta una estructura de grafo intervalo [4]. Este conjunto se propuso originalmente para el grafo huésped camino, y tiene la garantía de que existe una solución con cero errores en todas sus instancias. Sin embargo, esta propiedad no se ha demostrado formalmente para el grafo huésped ciclo. Por último, las instancias *Random* son grafos generados sin seguir un patrón estructural explícito. Cada uno de estos grupos de instancias cuenta con ciento diecisiete instancias, a excepción del grupo *Complete* que tiene ciento ocho. El número de vértices varía desde 10 hasta 250, mientras que el número de aristas va de 6 a 26281.

Para definir el conjunto preliminar de instancias, se ha utilizado un *software* de selección que, basándose en las propiedades de los grafos, elige las más distintas entre las instancias [10]. Las propiedades que se han empleado son el número total de aristas y vértices, el grado mínimo y máximo y el porcentaje de aristas negativas. En concreto, se han definido dos conjuntos de instancias: uno para la experimentación final, que contiene cuarenta y cinco instancias seleccionadas de las trescientas cuarenta y dos instancias totales; y otro subconjunto compuesto por diez instancias representativas a partir del conjunto de experimentación final, con el objetivo de acelerar el proceso de ajuste y experimentación preliminar.

## 4.2. Experimentación preliminar

Todos los experimentos se han ejecutado en el mismo entorno: una máquina virtual con una CPU AMD EPYC 7282 16-Core Processor y 32GB de RAM. Los algoritmos se han implementado en Java 21.

En el desarrollo esta propuesta algorítmica, se adoptó un proceso incremental, donde los diferentes componentes fueron agregados y validados de forma progresiva. El primer experimento realizado consiste en ajustar los parámetros del criterio voraz $g_3$, que cuenta con cuatro pesos $(w_1, w_2, w_3, w_4) \in [-1, 1]$. Para ajustar estos pesos, se ha utilizado *IRace* [8], un *software* de ajuste automático de parámetros. Los mejores valores obtenidos son: $w_1 = 1$, $w_2 = -0{,}5$, $w_3 = 0$ y $w_4 = 0$.

El segundo experimento se centra en los tres métodos constructivos presentados en la Sección 3.1: "Función Objetivo", que implementa el criterio voraz $g_1$; "Cliques", que utiliza el criterio $g_2$; y "Adyacencia", basado en $g_3$. Estos métodos constructivos, en su versión semivoraz, utilizan un parámetro $\alpha$, que regula la aleatoriedad de los métodos, que puede tomar valores en un rango de $[0, 1)$. Estos algoritmos se han limitado a un tiempo máximo de trescientos segundos, deteniendo la ejecución al alcanzar este límite y devolviendo la mejor solución encontrada hasta ese momento. Los resultados se muestran en la Figura 2a.

Los resultados de este primer experimento revelan que introducir aleatoriedad con el parámetro $\alpha$ no mejora el rendimiento de los constructivos, sino que tiende a empeorarlo. De hecho, las mejores soluciones se obtienen consistentemente con $\alpha = 0$, que simula el comportamiento voraz. Analizando individualmente, el método "Función Objetivo" empeora notablemente con aleatoriedad debido al coste computacional adicional que esta introduce. El método "Cliques" mantiene un rendimiento constante, ya que su criterio voraz genera candidatos muy similares, limitando el impacto de $\alpha$. Por último, el método "Adyacencia", muy efectivo con $\alpha = 0$, también se ve perjudicado al aumentar la aleatoriedad. Por lo tanto, para estos constructivos y este problema, la aleatoriedad controlada por $\alpha$ no aporta mejoras y puede ser contraproducente.

Ampliando el análisis, se estudió la dinámica temporal de los métodos constructivos, con el propósito de discernir cuál de ellos ofrece el mejor rendimiento a lo largo del tiempo. Para este análisis, se han ejecutado los algoritmos durante un total de quince minutos, evaluando la mejor solución encontrada por cada uno en intervalos de un minuto. Los resultados se representan en la Figura 2b.

El examen de estos resultados revela que el criterio $g_3$ emerge como el mejor método en términos de comportamiento temporal. No solo logra alcanzar las mejores soluciones, sino que lo hace en un tiempo significativamente menor en comparación con los demás. La efectividad de este constructivo radica en su estrategia de construcción de soluciones, que favorece el agrupamiento gradual de vértices con adyacencias positivas.

Adicionalmente, se llevó a cabo un experimento específico para cuantificar el tiempo promedio que cada método requiere para construir una solución. Tomando como referencia un constructivo puramente aleatorio, que requiere menos de un milisegundo por construcción, se compararon los tiempos de los métodos propuestos. El método más rápido resultó ser el método "Adyacencia", con un promedio de 1,06 milisegundos por construcción. A continuación, el método "Cliques" requirió un segundo por construcción, mientras que el método "Función Objetivo" demostró ser significativamente más lento que los demás, empleando aproximadamente tres veces más tiempo que el método 'Cliques'. Estos resultados evidencian la relativa lentitud de los métodos "Cliques" y "Función Objetivo", atribuible al considerable coste computacional inherente a la gestión de cliques y a la evaluación de la función objetivo en este problema, respectivamente.

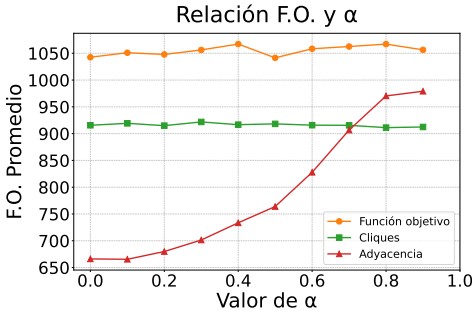

(a) Relación entre la función objetivo y el valor de $\alpha$ en los constructivos semivoraces.

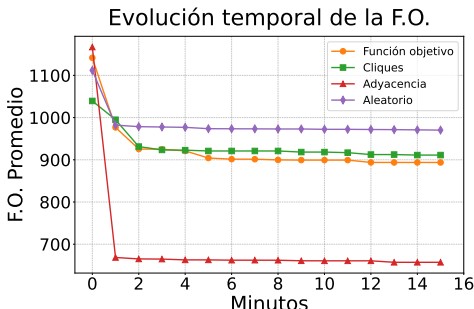

(b) Evolución de los métodos constructivos semivoraces aleatorio, función objetivo ($g_1$), cliques ($g_2$) y de pesos ($g_3$).

Como siguiente paso en el desarrollo del algoritmo, se incorporó una fase de búsqueda local a los métodos constructivos, integrándolos dentro de un esquema GRASP. Esta búsqueda local se basa en el movimiento de intercambio descrito en la Sección 3.2. Los resultados de esta integración, que muestran la relación entre el valor promedio de la función objetivo y el parámetro de aleatoriedad $\alpha$ en el contexto del esquema GRASP, se presentan gráficamente en la Figura 3.

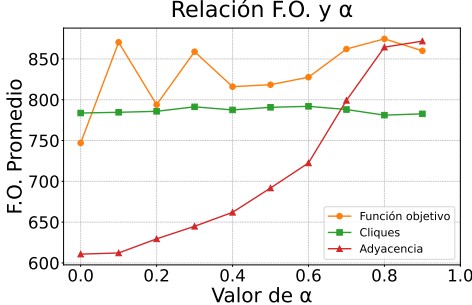

Figura 3: Evolución del método GRASP relacionado con el valor de $\alpha$.

En este experimento con GRASP, se observa que añadir aleatoriedad empeora los resultados. No es necesario añadir aleatoriedad extra con $\alpha$, ya que la aleatoriedad propia de los constructivos permite por sí misma diversificar la búsqueda lo suficiente. Por lo tanto, el método "Adyacencia" con $\alpha = 0$ destaca como el mejor algoritmo de esta propuesta. Los métodos constructivos de "Función Objetivo"

y "Cliques" mejoran un 2 % al aplicar la búsqueda local, mientras que el de "Adyacencia" mejora menos de un 0,33 %.

### 4.3. Comparativa final

Para la comparativa final se ha seleccionado la mejora variante de los algoritmos estudiados. En concreto, se ha demostrado experimentalmente que este es el algoritmo GRASP con el constructivo de "Adyacencia" con $\alpha = 0$ y con una búsqueda local de intercambios. Para medir la eficacia de esta propuesta algorítmica presentada se hace una comparativa con el BVNS [17] del estado del arte del CMinSA, que es el problema original del que surge el CMMSA estudiado en este trabajo. Para la comparativa, se adaptó el algoritmo BVNS original del estado del arte para CMinSA [17] al problema CMMSA, reemplazando únicamente la función objetivo, manteniendo el resto de la estructura del algoritmo inalterada. Para esta comparativa se han ejecutado ambos algoritmos para el conjunto de instancias final y se obtienen los resultados presentados en la Tabla 1. En concreto, se evalúan las siguientes métricas: promedio de la función objetivo (F.O.), promedio del tiempo de cómputo total en segundos (T. CPU) y número de mejores soluciones encontradas (# Mejores).

Tabla 1: Comparativa de los resultados obtenidos por el método constructivo "Adyacencia" con $\alpha = 0$ (Constructivo), el método constructivo con búsqueda local (GRASP) y el método BVNS [17].

| Método | F.O. | T. CPU | # Mejores (45) |
|---|---|---|---|
| Constructivo | 597,13 | 206,67 | 14 |
| GRASP | 556,93 | 208,18 | 45 |
| BVNS [17] | 700,18 | 344,54 | 1 |

Se puede observar que la propuesta GRASP obtiene resultados notablemente mejores, obteniendo todas las mejores soluciones. Comparado con el método de BVNS adaptado al CMMSA, este último solo obtiene una mejor solución, en la que obtiene el mismo resultado que el GRASP. Lo más notable del método de BVNS es el gran tiempo requerido para ejecutar que, debido al gran coste computacional del método constructivo, necesitando ciento treinta segundos más para ejecutar. Consecuentemente, el método GRASP es capaz de obtener mejores soluciones en un tiempo menor, y sin depender de aspectos teóricos específicos del grafo como pueden ser los cliques.

## 5. Conclusiones y trabajos futuros

En este trabajo se ha propuesto tres métodos constructivos voraces y sus correspondientes adaptaciones semivoraces para el problema de CMMSA, ejecutados en un esquema multiarranque. Entre los métodos voraces estudiados, ha destacado el método basado en la diferencia entre el número de adyacentes positivos asignados y sin asignar ($g_3$). Además de obtener los mejores resultados entre todos los métodos, este constructivo se distingue por su eficiencia computacional, requiriendo tres órdenes de magnitud menos tiempo para generar una solución en comparación con los otros métodos evaluados. En cuanto a los otros métodos constructivos, se ha comprobado que son considerablemente más costosos desde el punto de vista computacional, lo cual es especialmente notable en instancias de gran tamaño.

Por otro lado, la conversión de los métodos voraces a métodos semivoraces no ha resultado en una mejora de los resultados. Al contrario, a medida que se incrementaba la aleatoriedad en los métodos, los resultados empeoraban. Por lo tanto, se ha demostrado que, para este problema, los métodos voraces propuestos son más efectivos que sus respectivas adaptaciones semivoraces.

En la comparativa final, se ha contrastado la eficacia del algoritmo propuesto con el método del estado del arte para el CMinSA, obteniendo resultados iguales o superiores en todas las instancias evaluadas, y alcanzándolos en un tiempo menor. Por tanto, queda demostrado que este método multiarranque, gracias principalmente a su constructivo voraz $g_3$, constituye una aproximación más efectiva que otros métodos similares para el problema de CMMSA.

Como trabajos futuros, se propone estudiar otros métodos de búsqueda local como la vecindad de inserciones. Además, es importante plantear estrategias específicas para problemas basados en máximos, como puede ser por ejemplo el uso de criterios de desempate que, dadas dos soluciones con la misma función objetivo, determine cuál de las dos es más prometedora.

## Acknowledgments and Disclosure of Funding

Esta investigación ha sido parcialmente financiada mediante subvenciones: PID2021-125709OA-C22, financiado por MCIN/AEI/10.13039/501100011033 y por "ERDF A way of making Europe"; CIAICO/2021/224, financiado por Generalitat Valenciana; C1PREDOC24-047 y M2988 financiados por la Universidad Rey Juan Carlos 2022"; CIRMA-CM Ref. TEC-2024/COM-404 financiado por la Comunidad Autónoma de Madrid; TSI-100930-2023-3 (MCA07) financiado por Ministerio para la Transformación Digital y de la Función Pública; y "Cátedra de Innovación y Digitalización Empresarial entre Universidad Rey Juan Carlos y Second Episode" (Ref. ID MCA06).

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
