# OpenReview forum: "Heurísticas multiarranque para el problema de CMMSA"
_MAEB/2025/Congreso — MAEB 2025_

### Official Review · Reviewer_CDPL · 2025-03-17
**Heurísticas multiarranque para el problema de CMMSA**

**Rating:** 5
**Confidence:** 5

**Review:**

El artículo presenta un enfoque matheurístico innovador para el problema de CMMSA, combinando métodos constructivos heurísticos con la metaheurística GRASP. Los resultados experimentales son prometedores y están bien documentados, mostrando una mejora significativa en comparación con el estado del arte. El artículo está bien estructurado, con secciones claramente definidas que facilitan la comprensión del problema, la metodología y los resultados.
A continuación, se proponen algunos detalles menores que se proponen a los autores:
- Línea 282 página 7, el punto y coma se ha ido a la línea siguiente.
- Figura 2 a) poner el valor máximo del α en el eje x, es decir, en lugar de terminar en 0,8 debería terminar en 1. Importante, no pone Figura 2 en ningún lugar, solo pone a)
- Figura 2 b) en el eje x pone α , ¿son ejecuciones del algoritmo? ¿es tiempo? El valor de α va de 0 a 1 y no de 0 a 800 (o a 900). Por favor incluir el máximo y especificar qué se está mostrando en esa evolución. Importante, no pone Figura 2 en ningún lugar, solo pone b).
- Con respecto a la Figura 2 b) se indica en el texto que son intervalos de un minuto por eso entiendo que en el eje x ponga α . Entiendo entonces que se representa el tiempo, pero no en minutos sino en segundos y que en el eje x deberá ponerse que va de 0 a 900 segundos. Por favor, incluir el valor máximo en el eje x de 900 segundos y explicar bien.
- ¿Se podría acompañar la Figura 3 de una tabla que indique el porcentaje de mejora al aplicar la búsqueda local de cada método o al menos mencionar en el texto brevemente?
- Si el valor de α  que se selecciona es 0 con el método de adyacencia, usted está haciendo un algoritmo totalmente voraz. ¿Tiene sentido que salga ese valor de α ? Me parecería interesante ver qué ocurre considerando α uniforme y α random. Es decir, si el algoritmo se ejecuta 100 veces, α uniforme serían diferentes valores de α entre [0,1] dando un salto de 1/100, por ejemplo. De forma similar, α random considera que en cada ejecución del algoritmo se toma un α seleccionado aleatoriamente en el intervalo [0,1]
- Referencia 15 poner referencia completa y no poner et al.

---

### Official Review · Reviewer_yTUw · 2025-03-17
**El artículo presenta una variante novedosa del problema Sitting Arrangement  y el algoritmo propuesto para su resolución tiene un buen desempeño.**

**Rating:** 5
**Confidence:** 4

**Review:**

Este artículo aborda el problema Cyclic Min-Max Sitting Arrangement (CMMSA) y propone la aplicación de un algoritmo GRASP para su resolución. En la fase de construcción de soluciones, se presentan tres métodos. Los resultados obtenidos muestran que la mejor variante del GRASP propuesto supera en desempeño al algoritmo considerado como estado del arte.

En términos generales, el manuscrito está bien redactado y presenta el contenido de manera clara y estructurada.

A continuación, se plantean algunas observaciones y sugerencias de mejora:

1) En la línea 33 (pag 1), la definición de \sigma no es matemáticamente correcta. Debería definirse, o bien \sigma: E_G -> {-1,+1}, o bien indicar que \sigma({u,v}) \in {-1,+1}. Además, dado que el grafo no es dirigido, las aristas deben expresarse como {u,v} en lugar de (u,v).

2) Dado que los/as autores/as mencionan que uno de los principales aportes del artículo es la introducción del problema CMMSA, sería conveniente que su definición formal se presentara en una sección independiente, en lugar de incluirse dentro de la introducción. Asimismo, para comprender la relevancia de este trabajo, sería recomendable desarrollar con mayor profundidad la motivación detrás del estudio de este problema. En este sentido, se sugiere responder las siguientes preguntas: ¿Cuáles son las aplicaciones prácticas del problema? ¿Por qué es relevante su estudio?

3) En la sección 4.1, se menciona que el software de selección se basa en las propiedades de los grafos para elegir las instancias más distintas. Se debe especificar qué características se consideran para dicha selección.

---

### Official Review · Reviewer_1BsH · 2025-03-18
**Heurísticas multiarranque para el problema de CMMSA**

**Rating:** 4
**Confidence:** 4

**Review:**

This paper is about the Cyclic Min-Max Sitting Arrangement Problem and proposes various GRASP algorithms. The GRASP algorithms are tuned with Irace and the one with the heuristic function g_3 obtains the best result.

The article seems to be OK and I only have a few suggestions.

1) Why is the BVNS so bad when detailed to the GRASP. The GRASP has some improvements with irace, but it is unclear if BVNS has.

2) I think that you could more quickly say that the construction of good solutions has already some random decisions as can be seen by the results with alpha = 0.

3) Has the Irace been used with a training set different one by one with the test set? If not, maybe one should do it.

4) The writing in the figures should be larger by a factor of 2 to 3.

---

### Decision · Program_Chairs · 2025-03-20

Accept